# Challenge design roadmap

**Hugo Jair Escalante**                                    HUGO.JAIR@INAOEP.MX
*Instituto Nacional de Astrofísica, Óptica y Electrónica, Mexico*

**Isabelle Guyon**                                         GUYON@CHALEARN.ORG
*University Paris-Saclay, France, ChaLearn, USA, and Google, USA*

**Addison Howard**                                    ADDISONHOWARD@GOOGLE.COM
*KAGGLE, USA*

**Walter Reade**                                         INVERSION@GOOGLE.COM
*KAGGLE, USA*

**Sébastien Treguer**                                      STREGUER@GMAIL.COM
*INRIA, France*

**Reviewed on OpenReview:** *https://openreview.net/forum?id=Oc1taS2iYd*

## Abstract

This document serves as a comprehensive guide for designing and organizing effective challenges, particularly within the domains of machine learning and artificial intelligence. It provides detailed guidelines on every phase of the process, from conception and execution to post-challenge analysis. Challenges function as motivational mechanisms that drive participants to address significant tasks. Consequently, organizers must establish rules that fulfill objectives beyond mere participant engagement. These objectives include solving real-world problems, advancing scientific or technical fields, facilitating discoveries, educating the public, providing platforms for skill development, and recruiting new talent. The creation of a challenge is analogous to product development; it requires enthusiasm, rigorous testing, and aims to attract participants. The process commences with a comprehensive plan, such as a challenge proposal submitted for peer review at an international conference. This document presents guidelines for developing such a robust challenge plan, ensuring it is both engaging and impactful.

**Keywords:** Challenge design, organizer guidelines, challenge proposal

## 1 Before you start

This section delineates the essential inquiries that challenge organizers must address prior to initiating the process of challenge organization. Early consideration of these questions assists organizers in accurately estimating the resources required to achieve their objectives and in enhancing the preparation of their proposals. This document primarily focuses on data-driven challenges evaluated using quantitative objective metrics, with participants ranked on a leaderboard. Nevertheless, the methodology is also broadly applicable to jury-evaluated competitions and to benchmarks

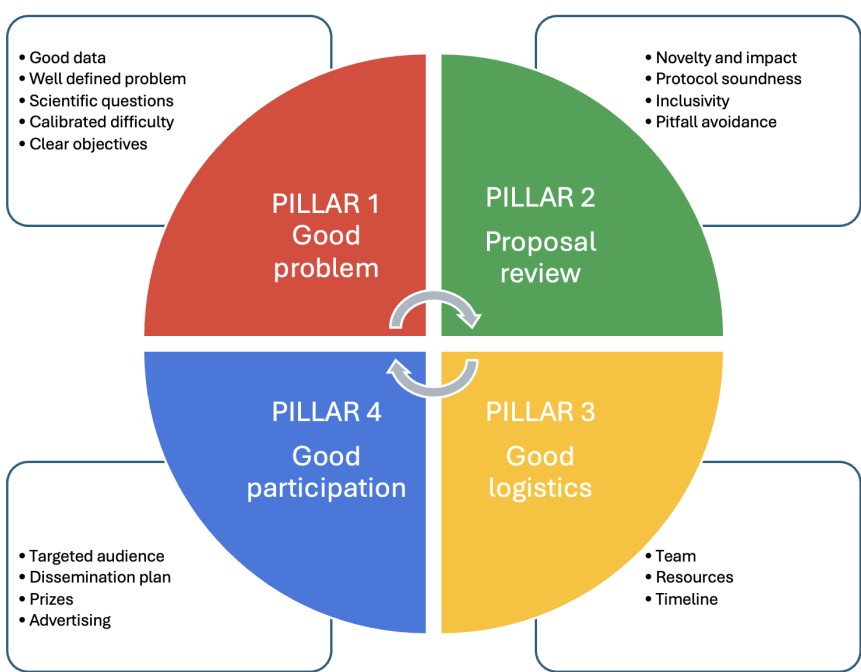

Figure 1: Challenge design principal pillars.

The material presented in the subsequent sections draws[1] on the preparation guidelines from organizations such as Kaggle[2], ChaLearn[3] and Tailor[4], as well as the NeurIPS proposal template, to which some of the authors contributed. This document constitutes Chapter 2 of the book "AI competitions and benchmarks: the science behind the contests[5]" (under preparation). The essential components of a successful challenge are summarized in Figure 1. Although the organization of this document does not strictly adhere to the structure of the figure, we will repeatedly refer to these essential ingredients.

## PILLAR 1: Good problem

This section highlights key prerequisites prospective challenge organizers must address. First of all, do you have a well defined challenge problem to be solved? Maybe not yet! You may just be interested in becoming a challenge organizer. In that case, we recommend that you **partner** with researchers, industrial or non-profit organizations who have data and an interesting problem to address. Even with a strong partner, this section may alert you to critical issues requiring attention.

---

1. Please note that while we have tried to provide context and justify every statement and recommendation of this document, part of the material presented in this document is based on the experience of authors who have dedicated a considerable amount of time to all aspects of challenge organization.
2. http://www.kaggle.com/
3. http://www.chalearn.org/
4. https://tailor-network.eu/
5. https://sites.google.com/chalearn.org/book/home

**Do you have (enough) good data?**

Data-driven challenges rely heavily on the availability of good data. Popular datasets can be beneficial for benchmarking purpose as they allow researchers to compare their results against a substantial body of prior work, providing context and establishing a standard for future research (Donoho, 2017). A dataset that has been used by many researchers in the past can be re-purposed to answer specific research questions. In Table 1 we list some **data sources** that can inspire you. However, using publicly available widely-used datasets carries a risk: competitors or the base models they use (such as pre-trained deep-net backbones) may have prior exposure to such data, which could bias evaluation or offer an unfair advantage.

Therefore, it is advisable to source datasets that have not yet been extensively explored by machine learning researchers. For further guidance on selecting, collecting, and preparing data, refer to document 3 of this book for a detailed description of the dataset development cycle (Egele et al., 2024). Additionally, Appendix C elaborates on data leakage, a major issue with data driven competitions.

The NeurIPS datasets and benchmarks track is also a growing resource of well-reviewed datasets. Using this resource type is particularly convenient prior to its publication. Otherwise, the dataset will have the public exposure, and means for data obfuscation or detecting prior exposure of models would be necessary. Please note that a considerable amount of datasets in this track are intended to become benchmarks (i.e., authors release everything you need to evaluate and compare models). Hence, this type of resource may not align well with certain challenge protocols, see Table 2.

When assessing potential data, organizers must consider numerous facets of **quality data**. Does the dataset contain biases (Ntoutsi et al., 2020) that would lead to an unacceptably biased model? Are there data leakages[6] that would spoil the objective of the challenge (Kaufman et al., 2012)? If the plan is to recycled or re-purpose data, do you have detailed information about the original intent (Koch et al., 2021)? Can you guarantee that the **test data is "fresh"**, i.e., that none of the participants had prior access to these data? If public re-purposed data is used for a challenge, is it **obfuscated enough** so that they are not recognisable to the participants, and/or hidden to the participants at test time? In addition to these considerations, it is critical that organizers ensure they have the **right to use the data and/or code** used in the challenge!

**Do you have a problem lending itself to a challenge?**

Having data is necessary, but not sufficient to organize a challenge. Do you have a **good definition of your problem** and have you tried yourself to solve it with some **simple baseline method**? Do you have a sense of how hard it is (it should **neither be trivial nor too hard to solve**)? If not, it is premature to organize a challenge with that problem, you need first to get familiar with this problem and be able to define criteria of success (called "metrics"), and have some preliminary ideas on how to optimize them. Make sure you understand how to **cast your problem into AI tasks**, which may range from machine learning tasks (binary classification, multi-class or multi-labels classification, regression, recommendation, policy optimization, etc.) (Burkov, 2019), optimization tasks (continuous, combinatorial or mixed; single- or multi-objective, etc.) (Aggarwal, 2020 - 2020), reasoning (logic or probabilistic), planning, constraint programming, . . . (Russell and Norvig, 2010), or a combination of several types of tasks. Having participated yourself to a challenge may be helpful. In

---

6. https://www.kaggle.com/docs/challenges#leakage

| Domain | Data source | Data type |
|---|---|---|
| Machine Learning | Kaggle datasets | The largest general-purpose ML dataset repository with >170K datasets in various formats, but generally coming with illustrative Jupyter notebooks. |
| CodaLab | CodaLab | A large repository with data from hundreds of challenges, mostly academic (Pavao et al., 2022). |
| Machine Learning | Hugging Face | More than 1000 datasets with well defined format/metadata and data loaders. Moatly for Audio, CV, and NLP modalities. |
| Machine Learning | OpenML | More than 5000 datasets, mostly in tabular format. |
| Machine Learning | UCI ML Repository | Historcal repository created in 1987 by D. Aha and his students. Hundreds of datasets, mostly small and in tabular format. |
| Reinforcement Learning | Farama gymnasium | New version of OpenAI gym. It includes a variety of environments, such as classic control problems and Atari games. |
| Miscellaneous | Data.gov | Over 200,000 datasets from the US government. The datasets cover a wide range of topics, from climate to crime. |
| Sensor Data | NOAA OpenSignal Array of Things) | Sensor data from a variety of sources, such as IoT devices, wearables, and environmental sensors, can provide rich information about the physical world. This data can be used to develop models for a wide range of applications, such as health monitoring, environmental sensing, and predictive maintenance. |
| Audio data | SpRUce Million Song VoxCeleb | Audio data, such as speech and music, is a rich source of information that can be used for a variety of applications, such as speech recognition, speaker identification, and music recommendation. |
| Textual data | Common Crawl Reuters News Amazon Reviews | Textual data, such as news articles, social media posts, and customer reviews, is a rich source of information that can be used for a variety of applications, such as sentiment analysis, topic modeling, and natural language understanding. |
| Satellite imagery | Planet Labs European Space Agency NASA | Satellite imagery can provide high-resolution images of the Earth's surface, which can be used for applications such as land use classification, urban planning, and disaster response. |
| Financial data | Quandl Yahoo Finance FRED | Financial data, such as stock prices, market trends, and economic indicators, can be used to develop models for predicting stock prices, identifying trading opportunities, and understanding economic trends. |
| Neuroscience Data | Open Neuro | A free and open platform for validating and sharing BIDS-compliant MRI PET MEG EEG iEEG |

Table 1: **Sources of data.** There is no good challenge without good data. An important aspect is to find "fresh data" to reduce the risk that the participants have been exposed to the challenge data previously, and can have an unfair advantage. Datasets commonly used in ML research (such as those colored in cyan) can be used as illustrative examples in the starting kit. Public datasets (such as those colored in yellow) can be used for the development phase (public leaderboard), but it is preferable to use novel fresh data for the final phase (private leaderboard).

that respect, the Kaggle book provides a gentle introduction to data challenges (Banachewicz and Massaron, 2022).

Additionally, not all problems lend themselves to a challenge. Importantly, challenges are "games of skills", NOT "games of chance". Can you devise a quantitative way of evaluating participants (using your metrics) in which **chance plays no role** (this is a legal requirement to organize scientific contest in most countries, to avoid that they fall under gambling regulations)? This may be particularly difficult. If the evaluation of participants entries relies on a statistic computed from data (typically some test set performance score using your metric), do you have **enough data** to obtain small error bars or a stable ranking of participants? In (Guyon et al., 1998), the authors provide a rule-of-thumb for classification problems. Usually, tens or thousands of examples must be reserved for test data, if you want good error bars; alternatively, you can use multiple datasets. Remember that, if the participants are allowed to make multiple submissions to the challenge and get feed-back on their performance on the test set on a leaderboard, they will soon overfit the leaderboard. In statistics, this is known as the problem of multiple testing. Leaderboard overfitting can be alleviated, to some extent, with a technique called "The ladder" (Blum and Hardt, 2015), which essentially boils down to quantizing the test set performance. However, to really prevent leaderboard overfitting, it is advisable to run the challenge in **multiple phases**: during a development phase, the participants are allowed to make multiple submissions and view their performances on a "public leaderboard"; during the final phase, the winners are chosen on the basis of a single submission, evaluated on a separate fresh test set. The performances are kept secret on a so-called "private leaderboard", invisible to the participants until the challenge is over. Since challenges are run in this way, leaderboard overfitting seems to have largely disappeared (Roelofs et al., 2019).

Be mindful that the **metric really reflects the objective you want to optimize**: a common mistake, for instance, is to use the error rate for classification problems, not distinguishing between the cost for false positive and false negative errors (e.g., it may be more detrimental to send a sick patient back home than to ask a sound patient to do one more medical exam). Also, would you rather your model provide you with a definitive prediction or a range of confidences? Finally, will you be declaring ties if performances between two participants are too close? See document 4 of this book for a detailed treatment on judging competitions, comprising critical aspects like the matching of metrics with objectives, statistical analysis evaluation, and fusion of multiple scores among other relevant topics.

**What are your scientific questions?**

What is the main problem[7] we want to address and would like to be solved? Asking the good questions is key to get results inline with the initial goals. What are the objectives of the challenge? Is our priority to address scientific questions, and which ones precisely, or to get as outcomes models easy to transfer to a production system with all its constraints in terms of robustness, explainability, performance monitoring, maintenance? Is the only objective, the final accuracy at the end of training without constraints on resources: compute, memory and/or time? Or should the participants also

---

7. Defining a problem and formulating scientific questions are closely related topics. However, they differ in their objectives and focus. When defining a problem one makes emphasis on finding well-defined tasks that can be solved by using machine learning, with clear goals and evaluation metrics. In contrast, scientific questions are broader and aim to advance the fundamental understanding of machine learning concepts, theories, and methodologies. They prioritize exploration, analysis, and theoretical contributions. While challenges focus on application and measurable outcomes, scientific questions emphasize knowledge generation and methodological innovation.

take into account limits in training time, compute power, memory size, and more, with the goal to find the sweet-spots for good trade-offs?

The definition of each task to achieve must help to solve a specific question raised by the challenge, but must also carefully take into account all constraints and reflections previously mentioned. Then what are the constraints in terms of data: volume, balance or unbalance of classes, fairness, privacy, external *vs.* internal, etc.? These questions are related to the tasks that are themselves related to the initial questions to be addressed. For each scientific question, you will then need to define some metrics allowing you to measure how well each participant answers the question: more details in Section 2 (*What metric for what purpose?*) of Document 4 of this book.

In general, AI challenges should have very specific objectives. There is a natural human tendency, when expending significant time and resources collecting and preparing data for a challenge, to want to answer as many questions as possible from the challenge. This is almost always counterproductive. While there may be considerable secondary information that can be gleaned at the conclusion of a challenge, challenges should be designed to have a very specific primary question to be addressed. This primary question is commonly in the form of the maximum predictive performance a model can extract from a given dataset.

Machine learning challenges often aim to address a multitude of scientific questions. These questions can be categorized using a taxonomy based on their overarching 5W themes: what, why, how, whether, and what for. Here's a potential breakdown:

1. **What (Discovery):** What patterns can be discovered in data? What features are more significant or relevant to the target variable? What groups or segments naturally form in the dataset? What are the characteristics of these clusters? An example of a discovery challenge would be the Higgs Boson challenge, that aimed at discovering a new high energy particle (Adam-Bourdarios et al., 2015).

2. **Why (Causality):** Why did a specific event or outcome occur? Are there variables that directly influence this outcome? Why does a certain data point deviate from the norm? For example, ChaLearn organized several causality challenges, including the Causation and Prediction challenge (Guyon et al., 2008) and the Cause-effect pair challenge (Guyon et al., 2019a).

3. **How (Prescriptive):** How can we allocate resources efficiently to achieve a goal? How can an agent take actions in an environment to maximize some notion of cumulative reward? For example the Black box optimization challenge asked participants to figure out how to optimize hyperparameters of models in a black box setting. The Learning to Run a Power Network (L2RPN) challenge series is asking how an agent can control the power grid to transport electricity efficiently, while maintaining equipment safe (Marot et al., 2020a,b, 2021).

4. **Whether (Comparative):** Whether Algorithm A is better than Algorithm B for a specific task? Whether a given preprocessing or hyperparameter setting X improves the model's performance over technique Y? Whether there is a trade-off e.g., between performance metrics (like precision and recall). Whether a model trained on dataset A performs better on dataset B compared to a model directly trained on dataset B (a transfer learning problem)? Whether a certain RL method performs better in environment condition X compared to condition Y?" For example, the Agnostic Learning vs. Prior Knowlege (AlvsPK) challenge answers the

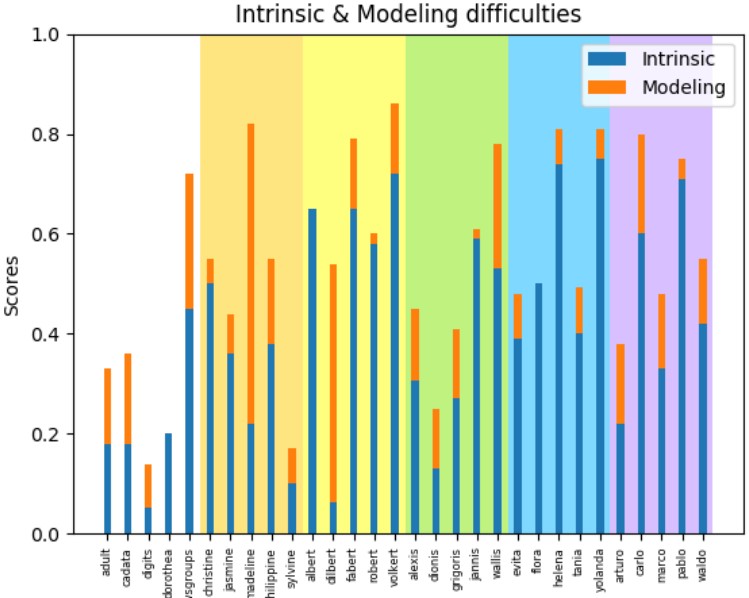

Figure 2: Example of intrinsic and modeling difficulty of datasets.

question whether prior knowledge is useful to devise a good preprocessing or whether using agnostic features is enough (Guyon et al., 2007).

5. **What For (Purpose-driven):** For whom might this model be unfair or biased? For what populations does this model perform sub-optimally? For what new tasks or domains can knowledge from one domain be useful? For example, the Jigsaw Unintended Bias in Toxicity Classification asked predicting and understanding toxicity in comments while considering potential biases against certain populations (Cjadams et al., 2019).

This taxonomy not only provides a structured way to think about scientific questions in machine learning but also helps in deciding the kind of algorithms, data processing techniques, and validation strategies that might be best suited to address them.

**Have you calibrated the difficulty of the challenge?**

Challenges that are overly difficult or too simple fail to advance the field. A critical aspect of challenge design is calibrating its difficulty, which may involve data engineering, metric refinement, and benchmarking tasks against established baseline methods.

Figure 2, taken from the AutoML challenge (Guyon et al., 2019b), shows how the difficulty of datasets might be calibrated. Test set scores are represented (normalized between 0 and 1, 0 is the level of random guessing). The height of the blue bar represents the score of the best model (our best estimated of the irreducible error of "intrinsic difficulty"). The height of the orange bar represents the range of scores between the best and the worst models, which we use to evaluate the "modeling difficulty". The datasets that are most suitable to separate methods well are those with small "intrinsic difficulty" and large "modeling difficulty". During beta-testing, you may want

to set up an "inverted challenge" among organizers, in which datasets are submitted against established baseline methods, then select those datasets with the largest ratio of modeling difficulty over intrinsic difficulty. If after adjusting the task difficulty, there is still little participation during the challenge, be ready to **lower the barrier of entry**, but providing more clues to get started, code snippets, notebooks, and/or tutorials.

**Do you have clear objectives?**

Are you more interested in finding a champion, benchmarking algorithms to evaluate or (incrementally) push the state of the art, or discovering brand new methods? Or are you simply interested in making your company/institution visible? While these three goals may be not mutually exclusive, your challenge design should take them into account.

In **recruiting challenges**, your goal is to find a **champion**. You may want to select a representative problem of what it is like to work at your company or institution to find top talents to employ. You will NOT need to put in effort in preprocessing data, designing a good API, etc.: the participants will be expected to do all the dirty work and show they **excel at solving all aspects of the problem**. Make part of the challenge deliverable that top ranking participants must deliver a technical report on their work to best evaluate them.

In **Research and Development challenges**, your goal is to benchmark algorithms. You may want to carefully design your API and have participants supply an object or a function, which addresses the **specific sub-task you have identified as a bottleneck of your problem**. Make sure to sort out licensing conditions of the winner's solutions. One simple way is to ask winners to open-source their code as a condition of being eligible for prizes.

In **Academic challenges**, your goal is to discover new methods for a problem that you largely do not know how to solve. You may want to have both quantitative and qualitative evaluations, e.g., in the form of a **best paper award**. This is because it is not obvious when you invent a new method to optimize it and get it to outperform others quantitatively, this may involve tuning and engineering.

In **Public Relation challenges**, your goal is to make your company or institution known, e.g., to attract new customers or students, and to expose your specific data or problem of interest to this public. It is essential to keep the challenge as simple as possible (at least in its problem statement) and as didactic as possible, and build around it great communication with the public, including using mass media. You may want to have intermediate goals and prizes and organize interviews of participants making good progress, to boost participation.

In **Branding challenges**, your goal is to put your name in front of a large community of data science practitioners by releasing a new technology or dataset (e.g., incentivizing the use of Tensorflow, or releasing a SOTA image classification set like ImageNet).

In **Hackathons** your goal is to provide innovative solutions to specific problems in a very short period of time. Participants are expected to work intensively and collaboratively around a theme, commonly aiming at open-ended innovation. You may want to dedicate extra effort in planning logistics (e.g., in terms of technical infrastructure and ensuring comfort for participants, providing mentorship and support, and even ensuring an enjoyable experience for participants).

**PILLAR 2: Proposal review**

Even experienced challenge organizers can make mistakes. Conferences with competition programs offer valuable opportunities to recruit participants and obtain feedback on your proposal.

We strongly recommend submitting your proposal to such venues. Other review instances may be required, such as ethical review boards, if human subjects are involved. Section 2 provides a proposal template, while this section highlights key questions to address for a strong and successful submission.

**Is your problem novel and impactful?**

Organizing a challenge entails significant responsibilities, as it engages the time and resources of both organizers and participants and may involve ethical considerations. Above all, reviewers will focus on the challenge's novelty and its potential impact (both positive and negative).

Ask yourself:

- whether other challenges or benchmarks have addressed the same problem before and whether the results or your new challenge will be **incremental or ground breaking**;

- whether you can illustrate your problem in one or several **domains of interest to the public**, which may include medicine, environmental science, economy, sociology, transport, arts, education;

- whether the outcome of the challenge will have a **practical societal or economical impact**;

- whether the approached task (or any aspect associated to it, e.g., application, domain, scenario) represents a potential hazard to users' rights, including **ethical issues**, e.g., linked to experimenting with human subjects, privacy concerns, etc.;

- how you can lower the barrier of entry to increase participation; this may mean e.g., using a uniform data format, providing data readers, providing sample code.

**Have you a well thought of protocol?**

After defining a challenge problem with robust data and metrics, numerous protocol choices remain. Will you use single or multiple metrics? Will these metrics span single or multiple phases or tracks? Will participants submit results or code? How much information will they have? What resources will organizers provide? Will participants face resource or time constraints?

Have you chosen your type of challenge protocol? Table 2 illustrates a hierarchy of challenge protocols for supervised learning tasks (Liu, 2021). As one progresses from one level to the next, participants have access to less information, whereas their submissions receive more. Level $\lambda$ is associated with challenges requiring result submissions, whereas the higher levels pertain to code submissions, from a class named ALGO. Level $\alpha$ is based on the premise of submitting pre-trained models. Level $\beta$ depicts a scenario of thorough code blind testing, where both training and testing occur on the platform. If multiple datasets are accessible, meta-testing can be conducted, given that participants are provided with examples of analogous tasks for meta-training. In the final stage, the $\gamma$ level, participants aren't provided with the meta-training set. Instead, they receive a basic task description. Consequently, from level $\lambda$ to $\gamma$, the submissions evolve to be increasingly autonomous, aligning more closely with genuine automated machine learning (AutoML). However, this also necessitates increased computational resources to implement the challenge.

Have you managed to reduce your challenge to a single clear objective that will give it focus? It is tempting, but generally unwise, to try to combine different metrics into a single objective.

Table 2: Hierarchy of challenge protocols.

| Level | Information available to participants | Information available to the algorithm only | Type of submission |
|---|---|---|---|
| $\lambda$ | Everything, except test labels | Nothing | `RESULTS` of test set predictions |
| $\alpha$ | Labeled TRAINING set | Unlabeled TEST set | `ALGO.predict()` |
| $\beta$ | META-TRAINING set | META-TEST set (each test set wo test labels) | `ALGO.fit()`, `ALGO.predict()` |
| $\gamma$ | Nothing, except starting kit and sample data | META-TRAINING set & META-TEST set (each test set wo test labels) | `ALGO.meta-fit()`, `ALGO.fit()`, `ALGO.predict()` |

When disparate scientific questions are force-fit into a single evaluation metric, the unintended consequences may be that the setup favors optimization of one question over the others, or that all the questions sub-optimized to optimize the combined metric. However, there are scenarios where multiple objectives genuinely arise, necessitating consideration of several metrics. For example, the metrics might include both accuracy and speed or memory footprint. In such cases, it is essential to strike a balance, ensuring that all objectives are addressed without overly compromising on any single aspect. Properly defined and carefully weighted metrics can help ensure that all objectives are optimized without undue sacrifices.

When facing multiple primary questions, it can be beneficial to introduce separate challenge tracks. For instance, the M5 forecasting challenge featured a track for precise point forecasts[8] and another for estimating the uncertainty distribution of the values[9]. Despite using the same dataset, the former adopted the weighted root mean squared scaled error, while the latter employed the weighted scaled pinball loss as their respective metrics. By creating distinct tracks, the challenge enabled researchers to advance the state-of-the-art in each area, rather than settling for methods that performed adequately across both but didn't excel in either. The choice boils down to whether the aim is to nurture specialist or generalist algorithms. To motivate participants to engage in multiple tracks while still promoting comprehensive methods, an incentivized prize structure can be adopted. For example, winners might receive a reward of $x$ for one track, double that amount for two, and exponentially more, as in $2^n x$, for triumphing in $n$ tracks.

There is a challenge format that does allow for more open-ended discovery of a dataset, which we'll refer to as an analytics challenge. The idea here is to provide data, give general guidance on the objective, and allow the competitors to analyze the data for new insights. An example of this was the NFL Punt Analytics challenge[10], where the goal was to analyze NFL game data and suggest rules to improve player safety during punt plays. While the scientific question was specific ("what rule changes would improve safety?"), the format allowed for a much broader exploration of the data and provided insights that wouldn't have surfaced by optimizing a single metric. While this can be beneficial, Analytics challenges tend to have much lower participation than predictive challenges, and require a significant amount of work after the challenge deadline to review and manually score (using a predefined rubric) each of the submissions from the teams.

Useful guidelines have been provided in Hutter (2019): "*The typical setup in machine learning challenges is to provide one or more datasets and a performance metric, leaving it entirely up to*

---

8. `https://www.kaggle.com/competitions/m5-forecasting-accuracy`

9. `https://www.kaggle.com/competitions/m5-forecasting-uncertainty`

10. `https://www.kaggle.com/competitions/NFL-Punt-Analytics-challenge`

*participants which approach to use, how to engineer better features, whether and how to pretrain models on related data, how to tune hyperparameters, how to combine multiple models in an ensemble, etc. The fact that work on each of these components often leads to substantial improvements has several consequences: (1) amongst several skilled teams, the one with the most manpower and engineering drive often wins; (2) it is often unclear* **why** *one entry performs better than another one; and (3) scientific insights remain limited. Based on my experience in both participating in several challenges and also organizing some, I will propose a new challenge design that instead emphasizes scientific insight by dividing the various ways in which teams could improve performance into (largely orthogonal) modular components, each of which defines its own challenge. E.g., one could run a challenge focusing only on effective hyperparameter tuning of a given pipeline (across private datasets). With the same code base and datasets, one could likewise run a challenge focusing only on finding better neural architectures, or only better preprocessing methods, or only a better training pipeline, or only better pre-training methods, etc. One could also run multiple of these challenges in parallel, hot-swapping better components found in one challenge into the other challenges. I will argue that the result would likely be substantially more valuable in terms of scientific insights than traditional challenges and may even lead to better final performance.*"

### Will your challenge be inclusive and favor open science?

Inclusivity and open science are often overlooked but should be prioritized early in challenge design. In fact, competition tracks in several conferences require that organizers justify the potential impact (open science, economical, societal, humanitarian, etc. ) of challenges, see Section 2. It is particularly important to consider the following aspects:

- **Encourage Open-Source Collaboration:** Highlight strategies to engage the open-source community, such as making challenge datasets and code repositories publicly available, fostering transparency, and encouraging contributions that improve the challenge infrastructure.

- **Support Underrepresented Researchers:** Offer specific guidelines or incentives to increase accessibility for underrepresented groups, such as cloud computing time, reduced registration fees in associate events, mentorship programs, or targeted outreach to underrepresented communities.

- **Promote Inclusive Participation:** Explore methods to create a more inclusive environment, like remote participation options, multilingual resources, and tailored support mechanisms to ensure broader and equitable engagement.

### Have you reviewed common pitfalls?

When everything appears ready, review this final checklist to ensure soundness.

1. **Lack of clarity.** Perhaps the most common mistake, is to offer a challenge that has a lack of clarity in the problem definition and the goals to be reached, a **too complex metric** defying intuition, or a **lack of focus** by addressing too many problems at once. When designing an AI challenge, it is important to understand that there is no way to optimize for all of the questions you might want to answer. It is better to put in the hard work up front to decide what the specific primary question should be and how to measure success with a single simple

metric. If there are secondary questions that you would like addressed, these should only be considered if they can be answered without jeopardizing the primary question. Clarity in the **rules** is also important. If you need a long list of rules for legal reasons, summarize them. Do not forget to have the participants accept the rules at the time of registration. See ChaLean contest rules, for inspiration. Add a FAQ page answering most frequently asked questions.

2. **Fatal flaws.** Another pitfall is to discourage serious competitors to enter because of **obvious flaws** (in data or in challenge rules). **Beta-test** your challenge thoroughly using volunteers (who will then not be eligible to enter the challenge) or solicit people you know well to enter the challenge as soon as it opens and report possible flaws. If possible, make a **"dry run"** or organize first a scaled-down version of your challenge to test its protocol, before you launch the "big one". Such trials will also allow you to **calibrate the difficulty of the task** to the target audience.

3. **Needless constraints.** Another way of discouraging participation is to put too many constraints or prerequisites to enter the challenge: having attended a previous challenge or event, registering with a nominative account, attending a conference, open-sourcing code, etc. While more constraints can be placed on the winners (or the entrants to the final phase), it is advisable to facilitate as much as possible entering the feed-back phase.

4. **Inconclusive results.** The success of the challenge also rests on having **quality data**, unknown yet to the public. We mentioned the problem of **bias in data** or **data leakage** in the introduction. Appendix C provides guidance on how to avoid falling into the most common traps when preparing data. In addition to having quality data, you must also have **sufficiently large datasets**. A common mistake is to reserve a fixed fraction of the dataset for training and testing (typically 10% of the data for testing), without anticipating the error bars. A simple rule-of-thumb to obtain at least one significant digit in a 1-sigma error bar, for classification problems, is to reserve at least $N = 100/E$ test examples, where $E$ is the anticipated error rate of the best classifier (the challenge winner) (Guyon et al., 1998). So, if you anticipate $E = 10\%, N = 1000$, if you anticipate $E = 1\%, N = 10000$.

## PILLAR 3: Good logistics

Organizing a challenge is akin to launching a product, requiring **managerial skills** in addition to technical expertise.

### Are you sufficiently qualified to organize your challenge?

There are many difficult aspects to tackle in the organization of a challenge and this can be daunting. It is rare that a single person is qualified to address them all. Think of **partnering with other experts** if you do not know how to answer any of the questions of the previous sections (or the following ones).

For instance, depending on the data types or modalities (tabular, univariate or multivariate time-series, image, video, text, speech, graph) and application-dependent considerations, **appropriate evaluation metrics** should be chosen to assess performance of the submissions. It may make a lot of difference if one chooses accuracy rather than balanced accuracy or AUC if classes are imbalanced, for instance. You may know that and know the difference between MSE and MAE for regression,

but do you know what SSIM, SHIFT, SURF are for image similarity? Do you know what F1 score is and what the difference is between micro-averaging and macro-averaging? Have you thought about whether you rather evaluate best objective value or time to reach given precision, for optimization tasks? Success or failure, time-to-solution for reasoning tasks? or do you need qualitative metrics possibly implying human evaluation (i.e., by some expert committee)? For details, see document 4 of this book.

Also, regarding data modalities, the choice of data and the evaluation of its quality require a lot of expertise. We have mentioned already the problem of data leakage, which leads to inadvertent disclosure of information about the "solution" of the challenge. Document 3 of this book reviews many more aspects of data that require attention, including legal aspects of ownership and attribution, privacy, fairness, and other legal aspects Egele et al. (2024). Each aspect may be better handled by an appropriate expert.

Another aspect requiring expertise will be the **preparation of baseline methods**. Make sure to include in your organizing team members who are knowledgeable of state-of-the-art methods and capable of implementing or running them on your tasks, to obtain baseline results. The code for baseline methods could be provided to the participants as part of a "starting kit". One motivating factor for the participant is "upskilling" themselves. Make sure you document well the baseline methods and provide good tutorial material, adapted to your audience. This will be much appreciated!

The adage "a rising tide lifts all boats" aptly fits this context. It conveys that when there's a general advancement or progress in a particular scenario (here referring to publicly accessible notebooks), all individuals involved reap the benefits, irrespective of their initial conditions. For instance, on Kaggle, you can kickstart a project using a pre-existing notebook created by someone else, paving the way for collective growth and assistance for all participants.

**Do you have enough resources to organize a good challenge?**

Challenges with **code submission** thought of being preferable to those with **result submission** in academia. This allows organizers to compare methods in a controlled environment and fosters fairer evaluations by providing to participants equal computational resources. However, with the advent of large foundational models (Chang et al., 2024), training on the challenge platform has become infeasible, for computational reasons. In that case, one can resort to letting the participants train their model on their own premises and submit the code of trained model, to be tested on the platform. Also, while code challenges are often better to raise equity across participants without similar access to resources, computational constraints can hamper participants from using the processing pipelines they are used to, which would not lead to establishing state-of-the-art performance. Other elements of fairness may include not requiring entry fees or the purchase of material (e.g., robots), not to favor entrants who are economically advantaged.

As a reminder, depending on the type of challenge protocol (Table 2), from level $\lambda$ to $\gamma$ are associated increasing computational resources to implement the challenge.

Do you have a **budget** to cover these costs and others (like preparing data)? Here is a non-exhaustive list of possible costs. See document 13 of this book for more details (Richard et al., 2024):

- Data collection, labeling, cleaning.

- Data preprocessing and formatting.

- Compensation of engineers preparing baseline methods or impementing the challenge.

- Computational resources to run the challenge.

- Prizes.

- Advertising.

- Organization of a workshop.

- Travel awards to attend the workshop.

- Fees for a challenge hosting service.

**Did you budget enough preparation and execution time?**

Depending on the novelty and difficulty of the challenge, it can take anywhere from a few weeks to a full year to prepare well a challenge. The participants also need sufficient time to familiarize themselves with the material and execute the tasks. In our experience, a minimum of 40 days is usually needed, but some challenges may require a few months. Finally, post-challenge analyses also require time and effort. Budgeting a few weeks in often necessary.

Did you budget enough time and account for possible delays in getting data, necessary protocol reviews or approvals?

**PILLAR 4: Good participation**

Few or no participants after months of organizing a challenge can be highly frustrating. While predicting a challenge's success is challenging, it is essential to take all possible measures to ensure strong and quality participation.

**Do you know your target audience?**

It is important to define the target audience, in order to design a challenge which is attractive enough and adapt the level of difficulty with a barrier to enter that is not too high.

Do you know the population of targeted participants? In order to adapt the difficulty level of a challenge, adapt the content and materials to enter in the challenge, it is recommended to define your target participants and ask yourself what are their backgrounds, skills, strength, weakness? Are they young students, experience professionals, research scientists, etc, with backgrounds in which fields?

If the target audience is a mix of beginners and more experienced practitioners in Artificial Intelligence, a crucial issue is to find a sweet spot, to set the barrier low enough to allow for beginners to enter without too much headache, while keeping the challenge challenging enough for experienced practitioners. Lowering the barrier to enter can be achieved by providing good documentation along with a simplified tutorial in a starting kit, providing compute resources to make it accessible to anyone, not only people with own access to farms of GPU or TPU. And at the same time, keeping the problem to solve interesting enough for experienced practitioners might require several levels of difficulty, several phases of the challenge.

Choose the start date, time length, and time investment required, according to the targeted audience. If you target researchers, make sure that being successful doesn't involve too much engineering time efforts with respect to the scientific contribution, and coordinate with other conferences, workshops and other challenges in the field.

Select a subject that aligns with the interests of your target audience at the time of the challenge. The suitability of a problem for a successful challenge is not solely determined by technical considerations but also by its potential to generate sufficient interest or avoid controversy. It is important to recognize that prizes constitute only a minor component of the incentive for participation, as most participants do not win, and the monetary rewards often pale in comparison to the time invested. Participants are, in effect, **donating their time**! driven by intellectual curiosity, professional growth, or community engagement.

When introducing the topic of your challenge, it is essential to craft a compelling "hook" that piques interest. Aim to attract a diverse range of participants. The beauty of this approach is that a machine learning expert, even without domain-specific knowledge, might clinch victory in a very specialized challenge. Conversely, an industry engineer looking to enhance their skills could very well triumph in a machine learning challenge.

**Do you have a plan to harvest the results of your challenge?**

Providing participants with opportunities to disseminate their work is both an important motivation to them to enter the challenge and a means of harvesting results. You may want to target one or several conferences (NeurIPS, KDD, WCCI, IDCAR have challenge programs, others welcome challenges organized in conjunction with workshops).

Inform participants of the challenge's start date well in advance to generate anticipation and allow adequate preparation. Ensure that sufficient time is allocated to finalize all necessary preparations before the launch. It is advisable to avoid scheduling challenge deadlines to coincide with major academic events, such as conference submission dates or student examination periods, as these may limit participation. Recurring challenges can foster a cumulative effect, gradually building a dedicated community that contributes to advancing the state-of-the-art over successive iterations.

Conferences do not usually have proceedings for their workshops, so you may have to make your own arrangements for proceedings. One venue that has been welcoming challenge proceedings in PMLR, the Proceedings of Machine Learning Research. The recently founded DMLR journal has also been welcoming challenge papers.

At the very least, top ranking participants should be asked to fill out fact sheets (see and example in Appendix B). Fact sheets can be a mix of textual descriptions providing a brief report in human readable format, and survey answers to a few questions, which can easily be aggregated as statistics. It is best to ask the participants to fill out the fact sheets before revealing the final leaderboard results, because otherwise non-winners have little incentive to put in the effort. Also, it is best to put as a condition to winning prizes to open-source the code and fill out the fact sheets.

Do not under-estimate the duration of challenge preparation, which, depending on the data readiness, the complexity of implementation of the challenge protocol and of establishing results may vary from a few days to over a year. Refer to document 3 of this book for recommendations on how to prepare data (Egele et al., 2024).

**Do you have monetary prizes?**

Publication venue is an essential motivation for participants of academic challenges. But, a recent analysis has determined that prizes are the greatest factor in boosting participation for Kaggle challenges. While overall participation is mostly driven by the approachability (a challenge with a tabular dataset will have more participation than one with 3D images), all else equal, the prize is 75 percent more important than any other factor. However, substantial prizes can attract participants more interested in monetary gain than scientific advancement, potentially leading to rule violations or the exploitation of challenge loopholes without disclosure. Another form of compensation is reimbursing travel expenses for attending a workshop where challenge results are discussed, which also facilitates result dissemination.

## 1.1 Do you have an advertising plan?

Last but not least, do not forget to **communicate well with your participants**. This starts with announcing your challenge ahead of time and advertising aggressively a few days into the challenge (once you are confident everything is running smoothly). Use all possible means available: mailing lists, social media, personal contact. Monitor the level of participation, get feed-back and stimulate participation, if needed, by adding bootcamps, webinars, and tutorial sessions. Make use of a forum and stimulate discussions between organizers and participants and between participants.

## 2 The proposal

In the Section, we provide a template of a proposal and provide a few tips about how to write a good proposal.

### ABSTRACT AND KEYWORDS

Briefly describe your challenge. Follow the following template (2 sentences maximum each topic):

- Background and motivation (stress impact).

- Tasks proposed and data used.

- Novelty (compared to previous challenges and benchmarks).

- Baseline methods and results (positioning the state of the art).

- Scientific outcomes expected (list questions asked).

Indicate whether this is a "regular challenge" running over a few months, a "hackathon" taking place over a day or two, and whether this will include a "live challenge" in the form of a demonstration requiring on-site presence. Also, provide up to five keywords, from generic to specific.

### challenge description

### BACKGROUND AND IMPACT

Provide some background on the problem approached by the challenge and fields of research involved. Describe the scope and indicate the anticipated impact of the challenge prepared (eco-

nomical, humanitarian, societal, etc.). Some venues privilege tasks of humanitarian and/or positive societal impact.

Justify the relevance of the problem to the targeted community and indicate whether it is of interest to a large audience or limited to a small number of domain experts (estimate the number of participants). A good consequence for a challenge is to learn something new by answering a scientific question or make a significant technical advance.

Describe typical real life scenarios and/or delivery vehicles for the challenge. This is particularly important for live challenges, but may also be relevant to regular challenges. For instance: what is the application setting, will you use a virtual or a game environment, what situation(s)/context(s) will participants/players/agents be facing?

Put special emphasis on relating the, necessarily simplified, task of the challenge to a real problem faced in industry or academia. If the task cannot be cast in those terms, provide a detailed hypothetical scenario and focus on relevance to the target audience.

Consider adding in a "hook" as an opening description, to attract those who are unfamiliar with the subject.

## NOVELTY

Have you heard about similar challenges in the past? If yes, describe the key differences. Indicate whether this is a completely new challenge, a challenge part of a series, eventually re-using old data.

## DATA

If the challenge uses an evaluation based on the analysis of data, provide detailed information about the available data and their annotations. Document your dataset thoroughly, using guidelines such as those provided in (Gebru et al., 2018). The data and their documentation should be ready prior to the official launch of the challenge.

*Quantity and quality of data:* Justify that: (1) you have access to large enough datasets to make the challenge interesting and draw conclusive results; (2) the data will be made freely available after the contest; (3) the ground truth has been kept confidential.

*Legal and ethical issues:* Verify and document permissions or licenses to use the chosen data. If new data are collected or generated, provide details on the procedure, including permissions to collect such data obtained by an ethics committee, if human subjects are involved. Minimize exposing personally identifiable information in datasets without informed consent and seek explicit consent when using real data from real people, explaining any inability to do so. If the data are recycled, verify that your dataset is not "deprecated". The authors of the original data may have recalled the dataset for some good reasons, e.g., data are biased in some way. The conference you are targeting may supply a list of deprecated datasets. Otherwise search on the Internet with your dataset name and "deprecated". For instance the search for "tiny images deprecated" yields this results: "The deprecation notice for Tiny Images was posted in direct response to a critique by external researchers, who showed that the dataset contained racist and misogynist slurs and other offensive terms, including labels such as *rape suspect* and *child molester*)".

See document 3 of this book for more details on how to prepare a good dataset (Egele et al., 2024).

TASKS AND APPLICATION SCENARIOS

Describe the tasks of the challenge and explain to which specific real-world scenario(s) they correspond to. If the challenge does not lend itself to real-world scenarios, provide a justification. Justify that the problem posed are scientifically or technically challenging but not impossible to solve. If data are used, think of illustrating the same scientific problem using several datasets from various application domains.

METRICS AND EVALUATION METHODS

For quantitative evaluations, select a scoring metric and justify that it effectively assesses the efficacy of solving the problem at hand. It should be possible to evaluate the results objectively. If no metrics are used, explain how the evaluation will be carried out. Explain how error bars will be computed and/or how the significance in performance difference between participants will be evaluated.

You can include subjective measures provided by human judges (particularly for live / demonstration challenges). In that case, describe the judging criteria, which must be as orthogonal as possible, sensible, and specific. Provide details on the judging protocol, especially how to break ties between judges. Explain how judges will be recruited and, if possible, give a tentative list of judges, justifying their qualifications. See document 4 of this book for help on evaluating a challenge.

BASELINES, CODE, AND MATERIAL PROVIDED

Describe baseline methods that can solve the problems posed in your challenge. Beta-test your challenge with such baseline methods and report the results. This is important to demonstrate that the challenge is not too easy nor too hard. You should have a range of baseline methods, from simple to sophisticated (state of the art methods). The results should show a large difference between unsophisticated and sophisticated methods.

Make the baseline methods part of the participants' "starting kit", which you should make publicly available together with sample data. The starting kit should allow participants to develop their solution and test it in conditions identical to those in which it will be tested on the challenge platform.

For certain challenges, material provided may include a hardware platform. Ideally the participants who cannot afford buying special hardware or do not have access to large computing resources should not be discriminated against. Find a way to make enough resources freely available to deserving participants in need (e.g. participant having demonstrated sufficient motivation by going through a screening test).

TUTORIAL AND DOCUMENTATION

Provide a reference to a white paper you wrote describing the problem and/or explain what tutorial material you will provide. This may include FAQs, Jupyter notebooks, videos, webinars, bootcamps.

**Organizational aspects**

PROTOCOL

Explain the procedure of the challenge:

- what the participants will have to do, what will be submitted (results or code), and the evaluation procedure;

- whether there will there be several phases;

- whether you will you use a challenge platform with online submissions and a leaderboard;

- what you will do for cheating detection and prevention;

- what you will do for beta-testing.

Code submission challenges can be resource-intensive but offer a plethora of benefits including:

- A controlled environment.

- Confidentiality of data.

- Equal time allocation for participants.

- Implementation of intricate protocols.

- Reduced chances of cheating.

- Accumulation of code for subsequent analysis.

RULES

In this section, provide:

1. A verbatim copy of (a draft of) the contest rules given to the contestants.

2. A discussion of those rules and how they lead to the desired outcome of your challenge.

3. A discussion about cheating prevention. Choose inclusive rules, which allow the broadest possible participation from the target audience.

It is imperative to clearly delineate the rules right from the outset and ensure they remain unchanged throughout. Maintaining transparency is key in fostering trust and participation engagement. Serious competitors prefer well-defined winning conditions. Evaluation procedures must be robust and tested prior to challenge launch to prevent issues. Although maintaining consistent rules is vital, organizers should retain the prerogative to amend rules or data if it's deemed essential. Such modifications, however infrequent, may be necessary to avert nullifying the entire challenge. Any alterations made early in the challenge are typically more acceptable to participants. Late-stage changes can cause discontent as participants might've dedicated significant time and resources, and such amendments might nullify their efforts. Organizers must balance the advantages of a change against its repercussions on the participants. For instance, last-minute minor data corrections might not merit the potential turmoil they could incite amongst competitors.

Organizers face numerous choices like:

- The option between single or multiple accounts.

- Anonymity regulations.

- Setting limits on submission counts.

- Deciding between result or code submissions.

- Instituting rebuttal or review mechanisms for results by fellow participants.

It's beneficial to have an adjudicating body or an uppermost appellate authority. The winners' codes should be subjected to internal result releases and peer review to ensure authenticity and merit.

We provide a concrete example of rules, corresponding to the challenge whose proposal is found in Appendix A.

### SCHEDULE AND READINESS

Provide a timeline for challenge preparation and for running the challenge itself. Propose a reasonable schedule leaving enough time for the organizers to prepare the event (a few months), enough time for the participants to develop their methods (e.g. 90 days), enough time for the organizers to review the entries, analyze and publish the results.

For live/demonstration challenges, indicate how much overall time you will need (we do not guarantee all challenges will get the time they request). Also provide a detailed schedule for the onsite contest. This schedule should at least include times for introduction talks/video presentations, demos by the contestants, and an award ceremony.

Will the participants need to prepare their contribution in advance (e.g. prepare a demonstration) and bring ready-made software and hardware to the challenge site? Or, on the contrary, can will they be provided with everything they need to enter the challenge on the day of the challenge? Do they need to register in advance? What can they expect to be available to them on the premises of the live challenge (tables, outlets, hardware, software and network connectivity)? What do they need to bring (multiple connectors, extension cords, etc.)?

Indicate what, at the time of writing this proposal, is already ready.

### CHALLENGE PROMOTION

Describe the plan that organizers have to promote participation in the challenge (e.g., mailing lists in which the call will be distributed, invited talks, etc.).

Also describe your plan for attracting participants of groups under-represented in challenge programs.

## Resources

### ORGANIZING TEAM

Provide a short biography of all team members, stressing their competence for their assignments in the challenge organization. Please note that diversity in the organizing team is encouraged, please elaborate on this aspect as well. Make sure to include: coordinators, data providers, platform administrators, baseline method providers, beta testers, and evaluators.

### RESOURCES PROVIDED BY ORGANIZERS, INCLUDING PRIZES

Describe your resources (computers, support staff, equipment, sponsors, and available prizes and travel awards).

For live/demonstration challenges, explain how much will be provided by the organizers (demo framework, software, hardware) and what the participants will need to contribute (laptop, phone, other hardware or software).

SUPPORT REQUESTED

Indicate the kind of support you need from the conference.

For live/demonstration challenges, indicate what you will need in order to run the live challenge.

## 3 A sample successful proposal

To exemplify the previous guidelines, we provide an example of successful NeurIPS proposal in Appendix A.

## 4 Conclusion

In this document, we have covered the fundamentals of organizing challenges. For a more comprehensive understanding of all aspects of challenge organization, please refer to the subsequent documents of this book.

The success of any challenge hinges predominantly on a strong team and a well-structured plan. A few key suggestions: don't underestimate the effort needed to execute your organization plan and bring in additional volunteers as necessary. Allow ample time to beta-test your challenge. If you're new to this, joining a seasoned organizing team to gain hands-on experience is likely the best approach.

## Acknowledgements:

This work was supported in part by ANR Chair of Artificial Intelligence HUMANIA ANR-19-CHIA-0022 and TAILOR EU Horizon 2020 grant 952215.

## Appendix A: Example of challenge proposal

This appendix provides an example of challenge proposal, the Cross-Domain Meta-Learning Challenge.

# Cross-Domain Meta-Learning
# NeurIPS 2022 Competition Proposal

**Dustin Carrión**[*]   **Ihsan Ullah**[*] **Sergio Escalera**   **Isabelle Guyon**   **Felix Mohr**   **Manh Hung Nguyen**

metadl@chalearn.org

## Abstract

Meta-learning aims to leverage the experience from previous tasks to solve new tasks using only little training data, train faster and/or get better performance. The proposed challenge focuses on "cross-domain meta-learning" for few-shot image classification using a novel "any-way" and "any-shot" setting. The goal is to meta-learn a good model that can quickly learn tasks from a variety of domains, with any number of classes also called "ways" (within the range 2-20) and any number of training examples per class also called "shots" (within the range 1-20). We carve such tasks from various "mother datasets" selected from diverse domains, such as healthcare, ecology, biology, manufacturing, and others. By using mother datasets from these practical domains, we aim to maximize the humanitarian and societal impact. The competition is with code submission, fully blind-tested on the CodaLab challenge platform. A single (final) submission will be evaluated during the final phase, using ten datasets previously unused by the meta-learning community. After the competition is over, it will remain active to be used as a long-lasting benchmark resource for research in this field. The scientific and technical motivations of this challenge include scalability, robustness to domain changes, and generalization ability to tasks (a.k.a. episodes) in different regimes (any-way any-shot).

**Keywords**

Deep Learning, AutoML, Few Shot Learning, Meta-Learning, Cross-Domain Meta-Learning.

## 1   Competition description

### 1.1   Background and impact

Traditionally, image classification has been tackled using deep learning methods whose performance relies on the availability of large amounts of data [1, 2]. Recent efforts in meta-learning [3, 4, 5] have contributed to making a lot of progress in few-shot learning for image classification problems. Tasks or "episodes" are made of a certain number of classes or "ways" and number of examples per class or "shots". Depending on the regime (number of shots and ways) various techniques have been proposed [6, 7, 8]. Despite progress made, allowing the community to reach accuracies in the high 90% in the last ChaLearn meta-learning challenge [9], evaluation protocols have a common drawback: Even when evaluated on multiple domains (e.g., insect classification, texture classification, satellite images, etc.), models meta-trained on a given domain are meta-tested on the same domain, usually simply assimilated to a large multi-class dataset from which tasks are carved. Furthermore, the number of ways and shots is usually fixed.

---

[*]The two first authors are co-lead organizers. Other authors are in alphabetical order of last name.

In contract, in this proposed challenge, domains vary, number of shots vary, and number of ways vary in every task (a.k.a. episode). For the purpose of this challenge, "domain" is not defined as a single large dataset, but as a collection of datasets from a similar application domain, with same type of images (object or texture) and same scale (microscopic, humans scale, or macroscopic). We call "mother dataset" a multi-class dataset from a particular domain from which we carve out tasks. Our mother datasets have each at least 20 classes and 40 examples per class (but usually many more). We have selected well differentiated domains, spanning object and texture recognition problems, at different scales. We formatted 30 mother datasets from 10 domains. Meta-training and meta-testing is performed on different mother datasets, one from each domain in each phase. For meta-testing, N-way k-shot tasks are drawn from one of the 10 mother datasets, with N in 2-20 and k in 1-20.

As documented in the literature, single domain meta-learning approaches have poor generalization ability to unrelated domains [10, 11, 12]. Nevertheless, this kind of generalization is crucial since there are scenarios where only one or two examples per class are available (e.g., rare birds or plants), and there is no close domain with enough information that can serve as source [13]. Therefore, addressing domain variations has become a research area of great interest.

In this sense, with the proposed challenge, we aim to provide a benchmark instrument that any person interested in the problem of cross-domain meta-learning for image recognition can use. Due to the rapidly increasing interest in meta-learning, we expect a large audience to be actively involved in this event. Additionally, the data we have collected for this challenge maximizes the societal impact by assembling datasets from various practical domains directly relevant to "AI for good", including medicine, ecology, biology, and others. We plan to reach out to a diverse community of participants with the organization of a bootcamp and prizes in several leagues (NewInML, Women, and participants of rarely represented countries). Moreover, the outcomes of this competition will help in the democratization of AI since the code of the winners will be open-sourced. On one hand, trained meta-learners can be used to create on-demand few-shot image classifiers for users having no particular knowledge of machine learning. On the other hand, meta-learners can be used to create few-shot model trainers, in other areas than image classification.

## 1.2 Novelty

Cross-domain meta-learning competition is part of the MetaDL competition series (see Table 1). It is built as an advancement of the NeurIPS 2021 MetaDL competition to target the problem of cross-domain meta-learning. Although the proposed competition is the third one around few-shot learning, the focus of the previous competitions (NeurIPS 2021 MetaDL and AAAI 2020 MetaDL-mini) was on single domain meta-learning, i.e., to build algorithms that perform well on unseen data, which is similar, but not same, to the seen data during training. On the other hand, the proposed Cross-domain meta-learning competition aims to tackle a problem more related to the real world scenario where the data can come from different domains. The idea is to build algorithms capable of generalizing the knowledge acquired during meta-training to any domain using only a few data. Thus, the models should "learn to learn" features that are domain independent to facilitate their adaptation to unseen tasks.

Table 1: ChaLearn Competition Series.

| Conference | Challenge | Description |
|---|---|---|
| ICML & NeurIPS 2016-18 | AutoML | Automating complete ML pipeline |
| WAIC 2019 | AutoNLP | Natural Language Processing |
| ECML PKDD 2019 | AutoCV | Computer Vision |
| ACML 2019 | AutoWeakly | Weakly Supervised Learning |
| WSDM 2019 | AutoSeries | Time Series |
| NeurIPS 2019 | AutoDL | Misc. domains |
| KDD cup 2020 | AutoGraph | Classification of Graph Data |
| InterSpeech 2020 | AutoSpeech | Speech Recognition |
| AAAI 2021 | 2020 MetaDL-mini | Few shot learning, trial run |
| NeurIPS 2021 | 2021 MetaDL | Few shot learning, meta-learning |

In the NeurIPS 2021 MetaDL competition (last in the series), algorithms were evaluated in each phase on tasks drawn from a single dataset of a given domain, experiments were repeated on multiple

datasets from various domains, and performances were fused for the final ranking. In contrast, in the proposed NeurIPS 2022 competition, the evaluation is carried out by pooling domains: tasks are drawn from any domain, both during meta-training and meta-testing. We have also increased the number of domains from 5 to 10 and the number of datasets from 15 to 30, see next section.

## 1.3 Data

Our competition will use a meta-dataset called Mata-Album[2], prepared in parallel with the competition, to be released after the competition ends. Several team members of the competition are also co-authors of this meta-dataset. It consists of 10 domains with three image "mother datasets" per domain. Although these datasets are publicly available, we selected for test purposes dataset that are not part of past meta-learning benchmarks. We preprocessed data in a standard format[3] suitable for few-shot learning. The preprocessing includes image resizing with anti-aliasing filters into a uniform shape of 128x128x3 pixels. The complete preprocessing pipeline is published[4].

Similar to our previous challenges, this competition has 3 phases: (1) **public phase** (release of starting kit and 10 public datasets), (2) **feedback phase** (participants submit their code to get instant feedback on 10 hidden datasets), and (3) **final phase** (the last submission of each participant from feedback phase evaluated on 10 new hidden datasets).

The datasets used for each phase are selected based on their novelty to the community: the most novel ones are used for meta-testing in the final phase. All the phases have one dataset per domain. Table 2 shows the datasets that will be used in each phase from each domain and their associated classification task. Except for one dataset, which might be replaced, we have obtained licenses for all the datasets. They will be released on OpenML [14] after the competition ends.

## 1.4 Tasks and application scenarios

In this challenge, we aim at pushing the full automation of few-shot learning by demanding participants to design learning agents capable of producing a trained classifier in the cross-domain few-shot setting. We will use the traditional *N*-way *k*-shot classification setting illustrated in Figure 1. This setting consists of three phases–meta-training, meta-validation, and meta-testing–which are used for meta-learning, hyperparameter tuning, and evaluation, respectively.

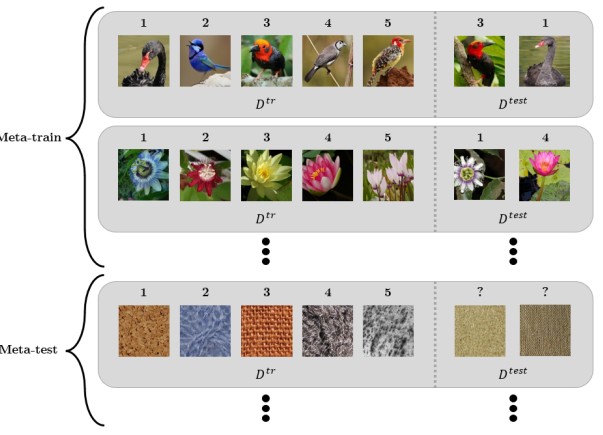

Figure 1: Illustration of 5-way 1-shot classification. This means that we have 5 classes and only one example of each class for learning. The test set includes a number of "query" examples, which are labeled in the meta-training set, but unlabeled in the meta-test set. Meta-validation tasks are not displayed. Figure adapted from [15].

Each phase is composed of multiple *episodes* which are small tasks $\mathcal{T}_j = \left( \mathcal{D}_{\mathcal{T}_j}^{train}, \mathcal{D}_{\mathcal{T}_j}^{test} \right)$ where $\mathcal{D}_{\mathcal{T}_j}^{train}$ and $\mathcal{D}_{\mathcal{T}_j}^{test}$ are known as *support set* and *query set*, respectively. Since the proposed setting assumes a cross-domain scenario, each task can be generated from any of the available mother datasets $\mathcal{D}$ in each phase. Moreover, the *N*-way *k*-shot classification setting states that every support

---

[2]Meta-Album: https://github.com/ihsaan-ullah/meta-album
[3]Data Format: https://github.com/ihsaan-ullah/meta-album/tree/master/DataFormat
[4]Preprocessing pipeline: https://github.com/ihsaan-ullah/meta-album/tree/master/PreProcessing

Table 2: **Datasets** to be used in the proposed competition. The 10 first are the "freshest" and will be used for the final test phase; the 10 in the middle will be used in the feedback phase; the last 10 datasets will be released in the public phase. The dataset written in red means that we have not obtained its license.

| Domain | Dataset | Meta-Album ID | Classification Task |
|---|---|---|---|
| **FINAL TEST PHASE** | | | |
| 1. Large Animals | Animal with Attributes | $LR\_AM.AWA$ | Mammals |
| 2. Small Animals | Insects | $SM\_AM.INS$ | Insects |
| 3. Plants | Fungi | $PLT.FNG$ | Fungi |
| 4. Plant Diseases | Plant Doc | $PLT\_DIS.PLT\_DOC$ | Sick and healthy leaves |
| 5. Microscopy | Kimia 24 | $MCR.KIMIA\_24$ | Human tissues |
| 6. Remote Sensing | RSD | $REM\_SEN.RSD$ | Satellite images |
| 7. Vehicles | Boats | $VCL.BTS$ | Boat types |
| 8. Manufacturing | Textures ALOT | $MNF.TEX\_ALOT$ | Textures |
| 9. Human Actions | MPII Human Pose | $HUM\_ACT.ACT\_410$ | Human pose |
| 10. OCR | OmniPrint-MD-6 | $OCR.MD\_6$ | Digital characters |
| **FEEDBACK PHASE** | | | |
| 1. Large Animals | Stanford Dogs | $LR\_AM.DOG$ | Dog breeds |
| 2. Small Animals | Insects | $SM\_AM.INS\_2$ | Insects |
| 3. Plants | PlantNet | $PLT.PLT\_NET$ | Plant types |
| 4. Plant Diseases | Medicinal Leaf | $PLT\_DIS.MED\_LF$ | Medicinal plants |
| 5. Microscopy | PanNuke | $MCR.PNU$ | Nuclei instance |
| 6. Remote Sensing | RSICB | $REM\_SEN.RSICB$ | Aerial images |
| 7. Vehicles | Airplanes | $VCL.APL$ | Airplane types |
| 8. Manufacturing | Textures DTD | $MNF.TEX\_DTD$ | Textures |
| 9. Human Actions | Stanford 40 Actions | $HUM\_ACT.ACT\_40$ | Human pose |
| 10. OCR | OmniPrint-MD-5-bis | $OCR.MD\_5\_BIS$ | Digital characters |
| **PUBLIC PHASE** | | | |
| 1. Large Animals | Birds | $LR\_AM.BRD$ | Bird species |
| 2. Small Animals | Plankton | $SM\_AM.PLK$ | Plankton types |
| 3. Plants | Flowers | $PLT.FLW$ | Flower categories |
| 4. Plant Diseases | Plant Village | $PLT\_DIS.PLT\_VIL$ | Plant leaves |
| 5. Microscopy | DiBas | $MCR.BCT$ | Bacterial colony |
| 6. Remote Sensing | RESISC | $REM\_SEN.RESISC$ | Aerial images |
| 7. Vehicles | Cars | $VCL.CRS$ | Car models |
| 8. Manufacturing | Textures | $MNF.TEX$ | Textures |
| 9. Human Actions | 73 sports | $HUM\_ACT.SPT$ | Human sports pose |
| 10. OCR | OmniPrint-MD-mix | $OCR.MD\_MIX$ | Digital characters |

set contains exactly $N$ classes with $k$ examples per class ($|\mathcal{D}_{\mathcal{T}_j}^{train}| = N \times k$). Furthermore, the classes in the query set $\mathcal{D}_{\mathcal{T}_j}^{test}$ must be present in the support set $\mathcal{D}_{\mathcal{T}_j}^{train}$ of a given task $\mathcal{T}_j$.

During the meta-training phase, the number of ways and shots for each task can be selected by each participant. However, during meta-validation and meta-testing, the number of ways will range from 2 to 20 with a number of shots ranging from 1 to 20, i.e., during meta-validation and meta-testing, the tasks will be any-way any-shot. To facilitate the creation of the learning agents, we will provide a large number of datasets formatted uniformly, amenable to meta-learning. Additionally, although different datasets are used in each phase, the domains remain the same.

The application scenarios are two-fold, corresponding to the 2 first prize leagues (see Section 2.4): (1) Few-shot image classification: a user seeks to create a classifier in a new domain, by providing a handful of examples of a number of classes. To that end, the meta-trained learning machine of the winners will be made readily available at the end of the challenge. (2) Meta-learning from limited amounts of meta-learning data: a user seeks to meta-train himself a learning machine in an application area other than image classification. To that end, the meta-learners of the winners will be open-sourced.

## 1.5 Metrics

Once the learning agents are meta-trained, they are presented with the meta-test dataset. The meta-test dataset consists of several episodes. For each episode, the agent is trained with the labeled support set $\mathcal{D}_{\mathcal{T}_j}^{train}$, and it is required to make predictions on the unlabeled query set $\mathcal{D}_{\mathcal{T}_j}^{test}$. The participants' performance on a domain will be the average classification accuracy of all tasks/episodes in the meta-test phase across all domains.

The error bars will be a 95% confidence interval of the mean classification accuracy computed as follows:

$$CI = \pm z^* \times \frac{\sigma}{\sqrt{n}}, \tag{1}$$

where $z^*$ is the corresponding value of the Normal distribution based on the confidence level, since in this case it is 95%, $z^* = 1.96$; $\sigma$ corresponds to the standard deviation of the accuracy obtained in all the episodes of the meta-test dataset, and $n$ is the number of episodes in the meta-test dataset. If computationally feasible, $n$ will be increased to obtain significant differences between top ranking participants at the 95% level. The stability of the ranking will also be evaluated by bootstrap resampling performances on the various tasks/episodes.

CI calculations and bootstrap experiments will only be indicative and not used to declare ties. Winners will be determined according to best rank in the final phase and ties broken according to first submission made.

## 1.6 Baselines, code, and material provided

The organizers will provide a "starting kit" that will be available to download directly from the challenge website as soon as the challenge starts. The starting kit will provide all the necessary code so the participants can make their local tests before submitting. Since 10 datasets will also be available, the starting kit will provide the corresponding data loader that will create the episodes as described in section 1.4. The competition is operationalized via a specific API to which participants must adhere, and which is documented in the starting kit.

The starting kit supplies five *baseline* agents that should be outperformed: (i) a random agent, which generates random predictions for each episode, (ii) a naïve approach, which accumulates all data from meta-train and trains a neural network on it and then applies it to the meta-test dataset, (iii) prototypical networks [16], (iv) MAML [17], and (v) prototypical networks with feature-wise transformations [13].

Table 3 shows the baseline results for the feedback phase that will be provided for the competition. This table includes the overall classification accuracy (i.e., the average classification accuracy of all meta-testing episodes) and the classification accuracy per dataset with their corresponding confidence intervals. The results of baseline (v) are not included in the table since it is under development.

## 1.7 Website, tutorial and documentation

We have set up a GitHub repository[5] for this competition which serves as the landing page and will be linked to the competition's CodaLab website. The GitHub repository covers the following points:

- Competition introduction and instructions for setting up the environment, including installing the required packages.
- Complete details of the evaluation process.
- Information about how to make a submission.
- Troubleshooting instructions for any possible issues and contact details for reporting issues.
- Link to CodaLab competition.
- Link to a dedicated forum on CodaLab platform for easy and efficient communication with participants.

In addition, a code tutorial is provided for the purpose of:

---

[5]GitHub repository: https://github.com/DustinCarrion/cd-metadl

Table 3: **Baseline results** for the feedback phase. TL and PN stand for Transfer Learning and Prototypical networks, respectively. The Overall row corresponds to the average classification accuracy of 1000 meta-testing episodes and its corresponding confidence interval. The remaining rows are the average classification accuracy and confidence interval of all the episodes carved out from each dataset. The name of the datasets corresponds to the Meta-Album IDs presented in Table 2.

|  | **Random** | **Naïve TL** | **PN** | **MAML** |
|---|---|---|---|---|
| Overall | $0.14 \pm 0.01$ | $0.16 \pm 0.01$ | $0.38 \pm 0.02$ | $0.17 \pm 0.01$ |
| *DOG* | $0.15 \pm 0.03$ | $0.10 \pm 0.02$ | $0.29 \pm 0.03$ | $0.17 \pm 0.02$ |
| *INS_2* | $0.14 \pm 0.03$ | $0.16 \pm 0.03$ | $0.31 \pm 0.04$ | $0.16 \pm 0.03$ |
| *PLT_NET* | $0.15 \pm 0.03$ | $0.17 \pm 0.03$ | $0.38 \pm 0.04$ | $0.18 \pm 0.03$ |
| *MED_LF* | $0.16 \pm 0.03$ | $0.16 \pm 0.03$ | $0.71 \pm 0.03$ | $0.27 \pm 0.04$ |
| *PNU* | $0.12 \pm 0.02$ | $0.16 \pm 0.03$ | $0.20 \pm 0.03$ | $0.17 \pm 0.02$ |
| *RSICB* | $0.12 \pm 0.02$ | $0.24 \pm 0.04$ | $0.72 \pm 0.03$ | $0.16 \pm 0.03$ |
| *APL* | $0.14 \pm 0.03$ | $0.12 \pm 0.02$ | $0.47 \pm 0.04$ | $0.16 \pm 0.03$ |
| *TEXT_DTD* | $0.13 \pm 0.03$ | $0.19 \pm 0.03$ | $0.30 \pm 0.03$ | $0.14 \pm 0.02$ |
| *ACT_40* | $0.15 \pm 0.03$ | $0.16 \pm 0.03$ | $0.21 \pm 0.03$ | $0.14 \pm 0.02$ |
| *MD_5_BIS* | $0.14 \pm 0.03$ | $0.15 \pm 0.03$ | $0.17 \pm 0.03$ | $0.16 \pm 0.03$ |

- Loading and discovering properties of data.
- Explaining the coding structure and the expected functions to be implemented in the code submissions.
- Providing instructions and examples for running the baseline methods on the public datasets.

## 2 Organizational aspects

### 2.1 Protocol

The competition will be hosted and run on the CodaLab platform[6] to which participants submit their solutions and receive summaries on their submissions. The 10 public datasets and the starting kit, which contains the API description and example agents can be downloaded from the GitHub repository. With this material, submissions can be drafted and tested (on the public datasets); no registration is required up to this point. To be able to make submissions to the system and hence enter the competition, the participants must create an account on the CodaLab platform, and then they can register for the competition. Neither the creation of the CodaLab account nor the registration into the competition has a fee. Once the participants are registered, they can submit agent solutions to the CodaLab server, which will immediately execute them on the feedback phase datasets and automatically display the results on the leader-board as soon as the run is finished.

As soon as the competition starts, the participants have direct or indirect access to 20 datasets in total. The leaderboard shown on the CodaLab platform, which is of main interest to the participants, is based on the 10 datasets of the feedback phase, to which the participants have no immediate access. The main benefit of the leaderboard is to enable a fair and objective evaluation of the submissions: all the submissions will be restricted by 10 GPU-hours of execution, and the computational resources will be the same, i.e., the CodaLab server will execute all submissions. The same hardware will be used in the final phase. However, the provision of this order of resources implies the necessity of limitations: Each participant will be allowed to make only 5 submissions per day and a maximum of 100 submissions in the course of the challenge. To allow the participants to perform other experiments on their own hardware, they can make use of the 10 datasets of the public phase, which are directly available through a download.

The proposed protocol was already tested in the previous challenges (2020 MetaDL-mini and 2021 MetaDL), but it was also tested when running the provided baselines. Furthermore, since we have team members of the CodaLab platform as co-applicants (Isabelle Guyon, Sergio Escalera) in the competition, we will be able to address CodaLab bugs and issues efficiently.

---

[6]CodaLab platform: https://codalab.lisn.upsaclay.fr

## 2.2 Rules

Draft of the rules:

- **General Terms**: This challenge is governed by the General ChaLearn Contest Rule Terms, the CodaLab Terms and Conditions, and the specific rules set forth.
- **Announcements**: To receive announcements and be informed of any change in rules, the participants must provide a valid email.
- **Conditions of participation**: Participation requires complying with the rules of the challenge. Prize eligibility is restricted by US government export regulations, see the General ChaLearn Contest Rule Terms. The organizers, sponsors, their students, close family members (parents, sibling, spouse or children) and household members, as well as any person having had access to the truth values or to any information about the data or the challenge design giving him (or her) an unfair advantage, are excluded from participation. A disqualified person may submit one or several entries in the challenge and request to have them evaluated, provided that they notify the organizers of their conflict of interest. If a disqualified person submits an entry, this entry will not be part of the final ranking and does not qualify for prizes. The participants should be aware that ChaLearn and the organizers reserve the right to evaluate for scientific purposes any entry made in the challenge, whether or not it qualifies for prizes.
- **Dissemination**: The challenge is part of the official selection of the NeurIPS 2022 conference. There will be publication opportunities for competition reports co-authored by organizers and participants.
- **Registration**: The participants must register to CodaLab and provide a valid email address. Teams must register only once and provide a group email, which is forwarded to all team members. Teams or solo participants registering multiple times to gain an advantage in the competition may be disqualified.
- **Anonymity**: The participants who do not present their results at the conference can elect to remain anonymous by using a pseudonym. Their results will be published on the leaderboard under that pseudonym, and their real name will remain confidential. However, the participants must disclose their real identity to the organizers to claim any prize they might win. See our privacy policy for details.
- **Submission method**: The results must be submitted through this CodaLab competition site. The number of submissions per day and maximum total computational time are restrained and subject to change, according to the number of participants. Using multiple accounts to increase the number of submissions in NOT permitted. In case of problem, send email to `metalearningchallenge@googlegroups.com`. The entries must be formatted as specified on the Instructions page.
- **Reproducibility**: The participant should make efforts to guarantee the reproducibility of their method (for example by fixing all random seeds involved). In the Final Phase, all submissions will be run three times, and the worst performance will be used for final ranking.
- **Prizes**: The three top ranking participants in the Final phase blind testing may qualify for prizes. The last valid submission in Feedback Phase will be automatically submitted to the Final Phase for final evaluation. The participant must fill out a fact sheet briefly describing their methods. There is no other publication requirement. The winners will be required to make their code publicly available under an OSI-approved license such as, for instance, Apache 2.0, MIT or BSD-like license, if they accept their prize, within a week of the deadline for submitting the final results. Entries exceeding the time budget will not qualify for prizes. In case of a tie, the prize will go to the participant who submitted his/her entry first. Non winners or entrants who decline their prize retain all their rights on their entries and are not obliged to publicly release their code.

**Discussion:** The rules have been designed with the criteria of *inclusiveness for all participants* and *openness of results* in mind. We aim to achieve inclusiveness for all participants by allowing them to enter anonymously and providing them cycles of computation (for the feedback phase and final phase) on our compute resources. This way, participants that do not have ample computing resources will not be limited by this and have a fair chance to win the challenge. We aim to achieve openness of results by requiring all participants to upload their base code and, afterward, fill in a fact sheet about the used methods. This allows us to conduct post-challenge analyzes on the winners' methods.

**Cheating prevention:** We will execute the submissions on our own compute cluster to prevent participants from cheating, and the testing datasets will remain hidden in the CodaLab platform.

Peeking at the final evaluation datasets will be impossible since those datasets are not even installed on the server during the feedback phase. Using the data in an unintended way during the final phase will be prevented by not revealing the true test labels to the agents ever, but only showing them to the scoring program on the platform. Moreover, different mother datasets for each domain are used in each phase to avoid both domain-specific cheating and also overfitting. We will also monitor submissions and reach out to participants with suspicious submission patterns. Finally, the candidate winners will have to open-source their code to claim their prize. Their code will be individually scrutinized by all other participants before they earn their prize.

## 2.3 Schedule and readiness

The competition preparations started in November 2021. The running duration of the competition will be 4 months, from June 2022 to September 2022. This time includes all the challenge phases. We are currently finishing the preparation of the baselines and working simultaneously on setting up the competition on the CodaLab platform. The data and protocol preparation is already finished. More details about the competition schedule are given in Table 4.

Table 4: Envisioned competition schedule.

| Date | Phase | Description |
|------|-------|-------------|
| November 2021 - January 2022 | Preparation | Data preparation |
| February 2022 - March 2022 | Preparation | Protocol preparation |
| April 2022 - May 2022 | Preparation | Baselines preparation and Setting up challenge environment |
| June 2022 | Public Phase | Start of the public phase, publicity |
| July 2022 - August 2022 | Feedbak Phase | Start of competition submissions |
| September 2022 | Final Phase | Evaluating performance on hidden datasets |
| October 2022 | Results | Notification of winners |

## 2.4 Competition promotion and incentives

To promote the competition, we will use the following channels:

- Mailing list from hundreds of participants from past challenges we organized.
- Advertisement on CodaLab, MLNews and comp.ai.neural-nets groups.
- Advertisement on the front page of OpenML (120,000 unique visitors yearly).
- In-network advertisement, e.g. personal Twitter accounts and personal emails.
- Organization of a bootcamp, held in presence at Universidad de La Sabana, Colombia, with remote participation permitted.

The bootcamp, organized similarly as the bootcamp of our previous meta-learning competition (https://metalearning.chalearn.org/metadlneurips2021) will encourage participation of South American students and researchers. Additionally, to encourage a diversity of participants and types of submissions, we will provide prizes in 5 different leagues:

- **Free-style league:** Submit a solution obeying basic challenge rules (pre-trained models allowed).
- **Meta-learning league:** Submit a solution that meta-learns from scratch (no pre-training allowed).
- **New-in-ML league:** Be a participant who has less than 10 ML publications, none of which ever accepted to the main track of a major conference.
- **Women league:** Special league to encourage women, since they rarely enter challenges.
- **Participant of a rarely represented country:** Be a participant of a group that is not in the top 10 most represented countries of Kaggle challenge participants[7].

The same participant can compete in several leagues. ChaLearn http://chalearn.org will donate a prize pool of 3000 USD, of which 600 USD will be distributed in each league ($1^{st}$ rank=300, $2^{nd}$ rank=200, $3^{rd}$ rank=100), and we will issue certificates to the winners. Furthermore, we will invite the winning participants to work on a post-challenge collaborative paper. We already have experience working on such a collaborative paper thanks to the analysis of the NeurIPS 2019 AutoDL challenge [18] and the NeurIPS 2021 MetaDL challenge [9].

---

[7]Kaggle ranking users: https://towardsdatascience.com/kaggle-around-the-world-ccea741b2de2

# A  Resources

This information does not count towards the 8 pages limit.

## A.1  Organizing team

- **Dustin Carrión**
  Université Paris-Saclay and LISN, France - dustin.carrion@gmail.com
  He is a second year master's student in Artificial Intelligence at Université Paris-Saclay, France. He is working under the supervision of Professors Isabelle Guyon and Sergio Escalera on cross-domain meta-learning. His research interests include meta-learning, self-supervised learning, semi-supervised learning and continual learning for computer vision applications. **He is the primary organizer with Ihsan Ullah. He contributed to the data collection, will implement the competition protocol, and will implement baseline methods.**

- **Ihsan Ullah** (https://ihsaan-ullah.github.io/)
  Université Paris-Saclay, France - ihsan2131@gmail.com
  He is a second year masters student in Artificial Intelligence at Université Paris-Saclay, France. He is working under the supervision of Professor Isabelle Guyon on challenge organization, data preparation, machine learning and meta-learning. **He coordinated the data collection and will contribute to the implementation of the competition protocol and the baseline methods.**

- **Sergio Escalera** (https://sergioescalera.com/)
  Full Professor at the Department of Mathematics and Informatics, Universitat de Barcelona, where he is the head of the Informatics degree. He is ICREA Academia. He leads the Human Pose Recovery and Behavior Analysis Group. He is an adjunct professor at Universitat Oberta de Catalunya and Dalhousie University, and Distinguished Professor at Aalborg University. He has been visiting professor at TU Delft and Aalborg Universities. He is a member of the Visual and Computational Learning consolidated research group of Catalonia. He is also a member of the Computer Vision Center at UAB, Mathematics Institute of the Universitat de Barcelona, and the Barcelona Graduate School of Mathematics. He is series editor of The Springer Series on Challenges in Machine Learning. He is vice-president of ChaLearn Challenges in Machine Learning, leading ChaLearn Looking at People events. He is co-creator of CodaLab open source platform for challenges organization and co-founder of the NeurIPS competition and Datasets & Benchmarks tracks. He is also Fellow of the ELLIS European Laboratory for Learning and Intelligent Systems working within the Human-centric Machine Learning program, member of the AAAC Association for the Advancement of Affective Computing, AERFAI Spanish Association on Pattern Recognition, ACIA Catalan Association of Artificial Intelligence, AEPIA Artificial Intelligence Spanish Association, INNS International Neural Network Society, Senior IEEE member, and vice-Chair of IAPR TC-12: Multimedia and visual information systems. He has different patents and registered models. He participated in several international funded projects and received an Amazon Research Award. He has published more than 300 research papers and participated in the organization of scientific events. He received a CVPR best paper award nominee and a CVPR outstanding reviewer award. He has been guest editor at TPAMI, JMLR, PR, TAC and IJCV, among others. He has been General co-Chair of FG20, Area Chair at CVPR, ECCV, NeurIPS, ECMLPKDD, AAAI, ICCV, WACV, IJCAI, FG, ICIAP, and BMVC, and Competition and Demo Chair at FG, NeurIPS, and ECMLPKDD, among others. His research interests include inclusive, transparent, and fair analysis of humans from visual and multi-modal data. **He is co-advisor of Dustin Carrion. He will provide expert advice on computer vision aspects and oversee fairness aspects of the competition.**

- **Isabelle Guyon** (https://guyon.chalearn.org/)
  Université Paris-Saclay, France and Chalearn, USA - guyon@chalearn.org
  She is chaired professor in artificial intelligence at the Université Paris-Saclay, specialised in statistical data analysis, pattern recognition and machine learning. She is co-founder and president of ChaLearn, a non-profit organisation dedicated to challenges in machine learning. Her areas of expertise include computer vision and bioinformatics. Prior to joining Paris-Saclay she worked as an independent consultant and was a researcher at ATT

Bell Laboratories, where she pioneered applications of neural networks to pen computer interfaces (with collaborators including Yann LeCun and Yoshua Bengio) and co-invented with Bernhard Boser and Vladimir Vapnik Support Vector Machines (SVM), which became a textbook machine learning method. She worked on early applications of Convolutional Neural Networks (CNN) to handwriting recognition in the 1990s. She is also the primary inventor of SVM-RFE, a variable selection technique based on SVM. The SVM-RFE paper has thousands of citations and is often used as a reference method against which new feature selection methods are benchmarked. She also authored a seminal paper on feature selection that received thousands of citations. She organised many challenges in Machine Learning since 2003 supported by the EU network Pascal2, NSF, and DARPA, with prizes sponsored by Microsoft, Google, Facebook, Amazon, Disney Research, and Texas Instrument. She was a leader in the organisation of the AutoML and AutoDL challenge series `https://autodl.chalearn.org/` and the meta-learning challenge series `https://metalearning.chalearn.org/`. **She is the advisor of the two primary organizers. She will oversee the good conduct of the competition, post-challenge analyses, and provide computational and platform resources.**

- **Felix Mohr** (`https://github.com/fmohr`)
  Associate Professor at Universidad de La Sabana, Chía, Colombia. He received his PhD in 2016 from Paderborn University, Germany, on the topic of automated service composition. He is an expert on Automated Machine Learning (AutoML) and is main author of several approaches and tools in this area including ML-Plan and Naive AutoML. His research interests include efficiency in stochastic optimization with a focus on the domain Automated Machine Learning. His latest interest is to use learning curves for meta-learning in order to increase the efficiency of model selection. **He will be in charge of the bootcamp organization. He will also provide expert advice on meta-learning and review protocols and rules of the competition.**

- **Manh Hung Nguyen** (`https://mhnguyenn.github.io/`)
  Chalearn, USA - `hungnm.vnu@gmail.com`
  He holds a Master's degree in Big Data Management and Analytics (BDMA) from the University of Paris-Saclay. He was an EMJM full-ride scholarship holder. His research interests include Automated Machine Learning, Meta-learning and Reinforcement Learning from both theoretical and practical points of view. He completed his research internship on Meta-learning at Inria under the supervision of Prof. Isabelle Guyon and Dr. Lisheng Sun-Hosoya. He was the lead organiser of the the IEEE WCCI 2022 Meta-learning from Learning Curves competition. **He will be in charge of communication and advertising. He will also contribute to data collection, implementation of the competition protocol and baseline methods.**

## A.2   Resources provided by organizers

We are relying on the following resources:

- **Competition infrastructure:** We will use the public instance of CodaLab `https://codalab.lisn.fr/` hosted by Université Paris-Saclay (the home institution of 4/6 organizers) as competition platform. To process participant submissions, we will supply 20 compute workers dedicated to the competition. Each compute worker will be equipped with one GPU NVIDIA RTX 2080Ti, 4 vCPUs and 16 GB DDR4 RAM. Our protocol was designed such that these computing resources should suffice to support the challenge.

- **Other computing Resources:** Through our sponsors, we have access to additional computing resources, which we used to prepare data and perform baseline experiments, and we will use in post-challenge experiments. We received a Google grant of 100,000 credits, equivalent to approximately 91575 GPU hours on a Tesla M60 GPU. Additionally, we have access to our institutional computation clusters.

- **Support Staff:** The CodaLab platform is administered by dedicated engineering staff at Université Paris-Saclay. During all the competition, the organizers will be available to support the participants through the forum of the challenge.

### A.3 Support requested

We will take the main responsibility for the publicity of our competition, ensuring plenty of participants from the NeurIPS community. To support us in this publicity, we count on the NeurIPS organization for the following matters:

- Display of the competition on the NeurIPS 2022 website.
- Referring participants to the various competitions that are organized.
- A time slot during the program, among which we can announce the winner and discuss the setup.

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

## Appendix B: Example of fact sheet

This appendix provides a template of fact sheet, used in the Cross-Domain Meta-Learning Challenge. The filled out fact sheets are found on the website of the challenge [11].

---

11. https://metalearning.chalearn.org/

# Cross-Domain MetaDL challenge fact sheet template

MetaDL organizing team

September 2022

This is the template for Cross-Domain MetaDL challenge[1] fact sheet. Please fill out the following sections carefully in a scientific writing style. Part of the filled fact sheet could be re-used as subsections in future publications.

Please edit this `.tex` file directly. Once finished, please zip the `.tex` file with all related files (e.g. generated PDF, `.bib` file) and send it to metalearningchallenge@googlegroups.com before **September 20, 2022**.

# 1 Team details

- **Team name \*:**

- **Team website URL (if any):**

- **Team leader name \*:**

- **Team affiliation \*:**

- **Team leader address \*:**

- **Team leader phone number \*:**

- **Team leader email \*:**

- **Name of other team members (if any)**

- **Username for Free-style league (if any):**

- **Username for Meta-learning league (if any):**

- **Publication list of each team member \*:**

- **Gender of each team member \*:**

- **Nationality of each team member \*:**

---

[1] https://codalab.lisn.upsaclay.fr/competitions/3627

# 2   Contribution details

- **Title of the contribution** *:

- **Summary** *
  In a few sentences outline what makes you proud of your contribution. Less than 50 words.

- **Motivation**
  Describe what motivates your method design and justify the importance of the approach. We expect a comparison with previous work and a clear explanation of the advantages of this approach to related work. Figures and tables may be added to support the text if needed.

- **Contributions** *
  An itemized list of your main contributions and critical element of success. **Highlight key words in bold.**
  Suggestions: Contrast your proposed method with others e.g. in terms of computational or implementation complexity, parallelism, memory cost, theoretical grounding.

- **Detailed method description** *
  In this part, contributions must be expanded and explained in more details. The explanations must be self-contained and one must be able to reproduce the approach by reading this section. You can explain and justify the approach by any means, e.g. citations, equations, tables, algorithms, platforms and code libraries utilised, etc. We expect a detailed explanation of the architecture, preprocessing, loss function, training details, hyper-parameters, etc.

- **Representative image / workflow diagram of the method** *
  An image (or several images) to support better description of your method. You can refer to these images in the method description part above.

- **Code repository** *
  Link to a code repository with complete and detailed instructions so that the results obtained on Codalab can be reproduced locally. This is recommended for all participants and mandatory for winners to claim their prizes.

# 3   Technical details

In this sections, multiple questions are asked on the technical details of your method. Please fill out this Google Forms *: https://forms.gle/3dasBFMwKCXv4kjx5

**REMEMBER**: After you filled out the form, you will get a link for later modification. Please right click on "Edit your response" and copy the link address for later modification and put it below:

_______________________________________________

# References

## Appendix C: Data leakage avoidance

**Leakage**

Leakage in machine learning refers to the inclusion of information during training that (a) is unavailable at prediction time for unseen data, and (b) artificially inflates model performance, thereby undermining generalizability. Leakage is often subtle, difficult to detect, and can severely compromise the integrity of machine learning competitions, sometimes rendering months of work unusable.

While competition organizers may sometimes be careless, leakage is often the result of its inherent complexity. Participants may also exploit leakage intentionally, driven by motivations such as monetary rewards or prestige, rather than advancing the field. Organizers must anticipate and address these issues proactively, as rules alone are insufficient deterrents. Preventing leakage requires careful preparation, including ensuring datasets are free from exploitable information and considering potential methods participants might use to gain an unfair advantage.

Leakage can occur in three main categories: access to ground truth, intrinsic properties of the data, and issues introduced during data processing. While these categories provide a framework for identifying leakage vectors, they are not exhaustive, emphasizing the importance of experience and diligence in designing robust competitions.

### ACCESS TO GROUND TRUTH

Ensuring the security of test set ground truth is critical for maintaining the integrity of machine learning competitions. Allowing participants to access these labels, even inadvertently, undermines the competition's validity. Obfuscating or making the data difficult to find is insufficient, as participants are often resourceful and capable of bypassing such measures.

Test data should be securely stored with access restricted to a limited, identifiable group of individuals who are excluded from participating in the competition. Additionally, care must be taken to prevent unintentional leakage, such as publishing graphs, descriptions, or prior content that reveals test labels. Credentials for accessing data should also not be publicly available, such as in repositories like GitHub. Proactive measures and thorough checks are essential to safeguard test data and prevent compromise.

### LEAKAGE INTRINSIC TO THE DATA

Leakage is a common issue in competition data, as it can arise in numerous subtle ways. For instance, in a challenge to classify images of cats and dogs, leakage might occur through timestamps if images of each category were collected at different times, or through metadata such as camera type or resolution, which may inadvertently predict the target. Even when metadata is stripped, features like image resolution or compression patterns can still reveal information, necessitating careful standardization of files.

A notable example of leakage involved a competition where a high score was achieved not by analyzing the data but by exploiting file attributes like size-on-disk and embedded timestamps. This underscores the importance of addressing both explicit and implicit leakage risks.

Leakage is especially challenging in fields like medical imaging, where variations in imaging equipment may introduce unavoidable biases. Organizers must carefully weigh the risks and benefits of including such data and decide which metadata to retain. Additionally, time series data poses

inherent leakage risks, as future predictions may inadvertently be influenced by temporal patterns in the training data.

Preliminary exploratory data analysis (EDA) and feature importance analysis can help identify potential leakage. However, organizers must ensure that any identified features are valid for use in future unseen predictions, as importance alone does not confirm leakage. Thorough consideration and proactive measures are essential to mitigate leakage risks effectively.

## LEAKAGE INTRODUCED WHEN PROCESSING THE DATA

Data leakage can be inadvertently introduced during the data processing phase through various mechanisms. One common source is ordering; if files are saved in a sequence that reflects the underlying labels or temporal patterns (e.g., all cat images saved first, followed by dog images), participants may infer the target variable indirectly. To mitigate this, files should be randomized in a repeatable, deterministic manner using set seeds or random-state parameters to ensure consistent reproducibility while breaking any unintentional ordering patterns. Similarly, leakage can occur when saving files with metadata or filenames that embed information about the target variable, such as timestamps, file sizes, or other distinguishing attributes. Ensuring that such metadata is stripped and filenames are anonymized is critical.

Additionally, the generation of synthetic data introduces unique challenges; if synthetic data retains traces of its generation process, such as embedding distinct features or artifacts that correlate with the target labels, it may provide unintended predictive signals. Careful preprocessing, validation, and testing of synthetic data are essential to prevent these artifacts from compromising the integrity of the competition. Proactively addressing these processing-related risks ensures fair competition and more generalizable model outcomes.

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
