# OpenReview forum: "Challenge design roadmap"
_DMLR — Accepted by DMLR_

### Review · Reviewer_dMWi · 2024-11-26

**Recommendation:** 3
**Confidence:** 2

**Summary Of Contributions:**

- This work provides a comprehensive set of guidelines for how to design a challenge in ML/AI (examples include things like the AutoML challenge, and the recent DataComp).
- This work breaks down the process into key questions in several areas, such as how to define the problem, how to organize the challenge, and how to incentivize participation. Within categories, taxonomies of protocols and subcategories are presented, and examples from previous challenges are provided.
- This work presents a template of a proposal that future organizers can use to develop their challenges. The work also provides an example NeurIPS challenge proposal that adheres to this template.

**Strengths:**

See strengths above.

**Audience:**

Yes

**Broader Impact Concerns:**

None.

**Claims And Evidence:**

Claims tend to lack evidence; see weakness 1 and 2.

**Datasets And Benchmarks:**

N/A

**Extended Submissions:**

N/A (book is under preparation)

**Limitations:**

See weaknesses above.

**Requested Changes:**

See weaknesses above. In particular, the following would secure my recommendation for acceptance:

1. Providing supporting evidence for all subjective statements in the work, or a disclaimer that all statements are views of the authors from their experiences organizing challenges (list and describe each challenge);
2. Providing more substructure (especially within section 1) and improving organization.
3. Integrating examples of more recent challenges, and increasing discussion around how the roadmap is constructed in light of recent shifts in AI research.

**Strengths And Weaknesses:**

**Strengths**
1. Extremely comprehensive discussion of many aspects of designing a challenge.
2. For certain aspects of the challenge, both pros and cons are weighed, while for other aspects, concrete, actionable recommendations are provided---this work has a definitive stance on how to construct challenges that may not be obvious to new organizers. For instance, the work strongly recommends that chance should play no role in the challenge (page 5), and a challenge should focus on improving a well-defined component of the ML pipeline; otherwise, insights may be limited (page 7).
3. Common pitfalls are discussed, so that potential organizers understand the historic consequences of a poorly designed challenge.
4. A template of a proposal is provided. This is very helpful for a challenge organizer, as they can fill out this template to get started and to ensure that they are generally answering all the important questions in designing a challenge.

**Weaknesses**

Content:
1. Many recommendations and statements do not appear to have supporting evidence. For instances "AI challenges should have very specific objectives" and "Analytics challenges tend to have much lower participation than predictive challenges."
2. While some statements are supported with citations or excerpts from others, it is overall unclear if the statements made in this work are based on the authors' own experiences, based on interviews/private discussion with other challenge organizers, or something else. The work should be very transparent regarding the process behind how the guidelines and key questions were constructed.
3. Minor: a recent type of challenge is the AI "hackathon", which often encourages open-ended development around some theme or base product. While hackathon challenges are briefly mentioned in the paper, it would be helpful to have a dedicated section of discussion for them, as hackathons do not always structure themselves in terms of conventional ML tasks and may be more open-ended than other challenges. An example that comes to mind is https://hackathon.bio/.
4. Minor: it would be helpful to draw on more recent challenges that reflect the shift in AI research towards larger models, inference-time improvements, etc. I do not hold this against the authors since this paper was probably in progress for a while before being submitted in June 2024. Some recent challenges include DataComp (https://www.datacomp.ai/dclm/) and BabyLM (https://babylm.github.io/)

Presentation:

5. The organization of this work could be improved. After I finished reading it, I felt that I had learned a lot about how to design a challenge, but I had trouble going back to particular parts of the paper to find specific guidelines. This could probably be improved by having more sections and subsections, as well as signposting at the beginning of each (sub)section. Moreover, a lot of content is expressed in a series of questions (for example, page 6 paragraph after "What are your scientific questions")---but chaining together many questions makes it difficult for the reader to go through them and identify if they are all of equal importance or serve as rhetorical followups to a single question.
6. Minor: While figure 1 is helpful, it does not seem to mirror the structure of the work.

---

### Review · Reviewer_TSJj · 2024-11-27

**Recommendation:** 3
**Confidence:** 3

**Summary Of Contributions:**

This position/guideline paper addresses the need for structured guidelines in designing challenges, which are increasingly used as gamified approaches to solve serious problems, promote scientific advancement, and educate the public.

Drawing from established preparation frameworks like those of Kaggle, ChaLearn, Tailor, and the NeurIPS proposal template, the paper provides actionable guidelines for designing robust and impactful challenges. These are useful and unique contributions to the general machine learning field.

It highlights...
- The complexity of creating effective competition rules that balance multiple objectives, such as engaging participants, solving real-world issues, and achieving scientific breakthroughs.
- The importance of developing a detailed competition proposal to be peer-reviewed at international conferences. While peer review does not guarantee quality, it compels organizers to consider their challenge's broader impact, address potential weaknesses, and enhance overall quality.

**Strengths:**

This position/guideline paper addresses the need for structured guidelines in designing challenges. I found this piece unique and actionable.

- **Structured Guidelines for Challenge Design**: It introduces actionable and practical frameworks for designing challenges, drawing from established resources like Kaggle, ChaLearn, Tailor, and the NeurIPS proposal template, tailored to the machine learning field.
- **Balancing Multifaceted Objectives**: It addresses the inherent complexity in creating competition rules that simultaneously engage participants, solve real-world problems, and drive scientific and technical advancements.
- **Promoting Rigorous Proposal Development**: It emphasizes the importance of crafting detailed competition proposals for peer review, which encourages organizers to enhance the quality, consider broader impacts, and identify potential weaknesses in their challenges.

**Audience:**

Yes

**Claims And Evidence:**

Yes

**Datasets And Benchmarks:**

N/A

**Extended Submissions:**

This is not an extended version but more of a condensed version of a book chapter. I am not sure as the can is not covered by eligibility criteria at https://data.mlr.press/acceptance-criteria.

**Limitations:**

The paper could be enhanced by addressing the inclusion of the open-source community and underrepresented researchers in both organizing and participating in challenges. Some points might include:
- **Encourage Open-Source Collaboration**: Highlight strategies to engage the open-source community, such as making challenge datasets and code repositories publicly available, fostering transparency, and encouraging contributions that improve the challenge infrastructure.
- **Support Underrepresented Researchers**: Offer specific guidelines or incentives to increase accessibility for underrepresented groups, such as reduced registration fees, mentorship programs, or targeted outreach to underrepresented communities.
- **Promote Inclusive Participation**: Explore methods to create a more inclusive environment, like remote participation options, multilingual resources, and tailored support mechanisms to ensure broader and equitable engagement.

**Requested Changes:**

- **Tables and Figures**: Table 1 has text that extends beyond borders, the images should be in PDF formats.
- **Inconsistent Formality**: Certain parts of the paper adopt an informal tone that detracts from its scholarly nature. Examples like "they are largely donating their time!" (page 10) and "Let your participants know in advance" (page 12) could be rephrased for a more professional tone, aligning with the expectations of an academic audience. These are just examples and such informal wording can be found throughout this paper.

**Strengths And Weaknesses:**

Strengths: See below.

Weaknesses:
- **Formatting Issues**: Table 1 has text that extends beyond borders, making it hard to read and unprofessional. This should be corrected to ensure clarity and a polished presentation.
- **Lack of Detail in References**: Referring readers to another chapter without summarizing key points leaves gaps in understanding. A brief overview of the referenced content would enhance the document’s self-sufficiency and readability.
- **Inconsistent Formality**: Certain parts of the paper adopt an informal tone that detracts from its scholarly nature. Examples like "they are largely donating their time!" and "Let your participants know in advance" could be rephrased for a more professional tone, aligning with the expectations of an academic audience.

---

### Review · Reviewer_pri2 · 2024-12-21

**Recommendation:** 3
**Confidence:** 3

**Summary Of Contributions:**

This paper is a chapter of a book talking about how to host a AI/ML competition. The paper can serve as a guidance for organizers, including the dataset, benchmark, evaluation metrics, the preparations of proposal, target audience, etc.

**Strengths:**

See above

**Audience:**

Yes

**Claims And Evidence:**

N/A

**Datasets And Benchmarks:**

N/A

**Extended Submissions:**

No

**Limitations:**

See above

**Requested Changes:**

See weakness

**Strengths And Weaknesses:**

Strengths:
- The content is comprehensive and systematic. As an experienced AI/ML competition organizer, I have learned a lot from reading  it, which I believe the paper is of importance and significance to guiding the organization of competitions.
- The writing flow is overall easy to follow and lots of examples are involved to show the opinions, making the paper friendly to read.
- The proposal part is more technical and the template example attached in the appendix is appropriate and helpful.

Weaknesses and limitations:
- Even if Figure 1 is intuitive and clear, it is not cross-referred in the following text, which weakens the coherence of the paper. It needs more explanation of Figure 1 in the beginning and more callbacks in the follow detailed sections.
- The difference between “a problem leading to a challenge” and “scientific questions” is not that clear. If I understand it correctly, the former is to elicit the choice of dataset and benchmark while the goal of the latter is to show the importance of metrics. However, the introduction or definition of “metric”has already been mentioned in the first paragraph on page 5 before “scientific questions” part, which makes the organization a bit confusing.
- Speaking of choice of dataset and benchmark for competition, it is better to discuss if any new dataset and benchmark introduced in the paper (e.g. NeurIPS dataset and benchmark track) should be proper to host a challenge. If not, show the justification.
- Some minors: Table 1 is too large and some of words in Domain column are truncated. It is recommended to use smaller font. In the last paragraph on page 7, it should be with a clear reference or citation when quoting other’s sentences.